# Enhancing reproducibility in stable isotope analysis (SIA) of fish eye lenses: A comparison between lamina number and diameter

Alexandra Chu[1], Danhong Ally Li[1]*, Miranda Bell-Tilcock[2], Miranda Lowe-Webb[1], Carson Jeffres[1], Rachel C. Johnson[1,3]

1 Center for Watershed Sciences, University of California, Davis, California, United States of America, 2 Delta Stewardship Council, Sacramento, California, United States of America, 3 National Marine Fisheries Service, Southwest Fisheries Science Center, Santa Cruz, California, United States of America

☯ These authors contributed equally to this work.

* dhli@ucdavis.edu

## Abstract

Analyzing stable isotopes in archival tissues, such as fish eye lenses, is used to document shifts in feeding ecology, diet, habitat use, and to reconstruct life history. Fish eye lenses grow throughout their ontogeny, forming multiple sequential layers, or laminae. These laminae represent the chronology of the fish's life, much like tree rings, which record environmental conditions over time. Lenses are protein-rich, which makes them an ideal structure for analyzing light isotopes such as $\delta^{13}C$, $\delta^{15}N$, and $\delta^{34}S$. These light isotopes are primarily integrated into the lens tissue through the fish's diet, where they are bound to amino acid structures during protein synthesis. As research begins to emerge using eye lenses to reconstruct the life histories of fishes, the need for a reproducible method of delamination grows. For this study, each researcher independently delaminated one lens from each of the 10 adult Chinook Salmon (*Oncorhynchus tshawytscha*). Lens lamina number, diameter (mm), and mass (mg) of each lamina were recorded. Laminae were then submitted for stable isotope analysis of both $\delta^{13}C$ and $\delta^{15}N$. Isotope values were used as a validation to compare delamination patterns between researchers. $\delta^{13}C$ and $\delta^{15}N$ values from the lenses were then plotted using both the assigned lamina number and lens diameter to compare the difference between researchers. Analysis based on lamina number showed significant shifts in isotope values and variability in lamina counts between researchers. However, when lens diameter was used instead of lamina number, isotope patterns throughout the lenses of the same fish were nearly identical. Using lens diameter removes subjectivity between researchers, thereby increasing the reproducibility of the technique and providing a more robust interpretation of the data.

**Data availability statement:** All collection and isotope data used in this paper are published on Dryad and accessible via DOI: https://doi.org/10.5061/dryad.8cz8w9h1s

**Funding:** Funding for this project was provided by California Department of Fish and Wildlife Proposition 1 agreement number P1896030. PIs: Carson Jeffres and Rachel Johnson. CDFW provided some of the tissue samples used in this study but had no role in study design, data analysis, decision to publish, or preparation of the manuscript.

**Competing interests:** NO authors have competing interests

## Introduction

Understanding animal diets is fundamental to ecology, not only for unraveling the ecological roles of species but also for identifying key regions and habitats that support biodiversity. Dietary analysis provides insights into food web dynamics, predator-prey relationships, and ecosystem health [1,2]. Investigating dietary shifts is particularly important for species with complex life cycles that rely on different food sources at distinct developmental stages [3]. However, capturing a complete dietary history is challenging, especially for migratory species, as traditional methods like prey sampling and gut content analysis offer only a snapshot of an individual's diet at a specific time [4].

Fish eye lenses are an emerging archival tissue type that allows researchers to trace diet history throughout an individual's life [5]. Eye lenses consist of three distinct zones: the gelatinous living outer cortex, the hardened intermediate zone and the dense inner core [6,7] (Fig 1). As the fish matures, each lamina becomes metabolically inert after its formation, preserving the isotopic composition of the fish's diet during that time (Fig 2A) [8]. This layered structure makes it the ideal tool for tracing dietary information throughout an individual's life [8] (Fig 2B).

Stable isotope analysis (SIA) of fish eye lenses has been conducted across different species, with studies investigating ontogenetic shifts, resource partitioning, and trophic histories [2,7,9]. SIA provides a powerful tool for studying animal diet histories, movement patterns, and natal origins without requiring the subsequent recapture of the same individual [10–12]. This method has been successfully used to create isoscapes, or spatial maps that depict unique isotopic values across the landscape [2,10]. Stable isotope analysis of recovered tissues is especially useful when studying elusive species that are difficult to tag. For example, SIA of eye lenses revealed habitat shifts in Atlantic goliath grouper (*Epinephelus itajara*), emphasizing the role of mangrove connectivity in conservation [1]. Similarly, by combining SIA of eye lenses with a mixing model, researchers have discovered that Clear Lake Hitch (*Lavinia exilicauda chi*) transition from benthic to pelagic food sources, pinpointing the specific size at which this transition occurs [5]. In addition to fishes, $\delta^{13}C$ and $\delta^{15}N$ laminae values of neon flying squid (*Ommastrephes bartramii*) were used to compare dietary shifts across the life stages in both eastern and western stocks [13].

In previous stable isotope analyses of fish eye lenses, researchers have indexed sampling position by counting laminae [2,8] or measuring lens diameter [4,5,8,9]. Counting laminae is appealing because it requires no specialized measuring equipment. Thus, lamina number is a more accessible metric to use compared to lens diameter. Measuring the lens, however, necessitates specialized and often costly equipment like microscope cameras and imaging systems. Simpler tools, such as ocular micrometers [4,8] or digital/ Vernier calipers [13], can suffice but complicate workflow and sacrifice precision.

Using lamina number as an indexing metric introduces bias. Each researcher must decide where one lamina ends and the next begins, thus counts can vary systematically from person to person. When multiple researchers process the same collection of lenses, those subjective differences can translate into inconsistent

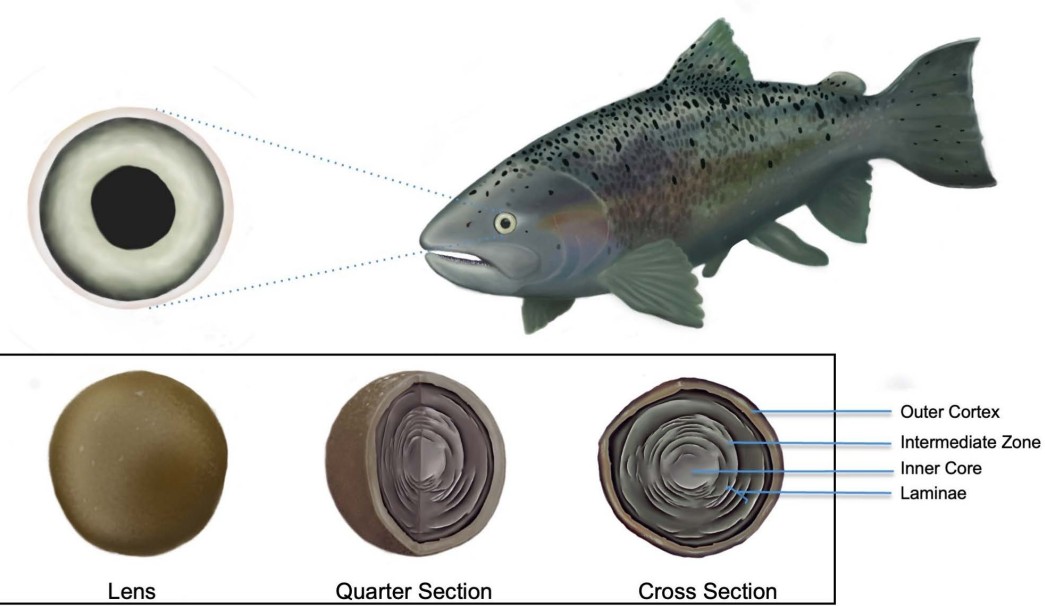

**Fig 1. Illustrations of Chinook Salmon (*Oncorhynchus tshawytscha*) eye lenses.**

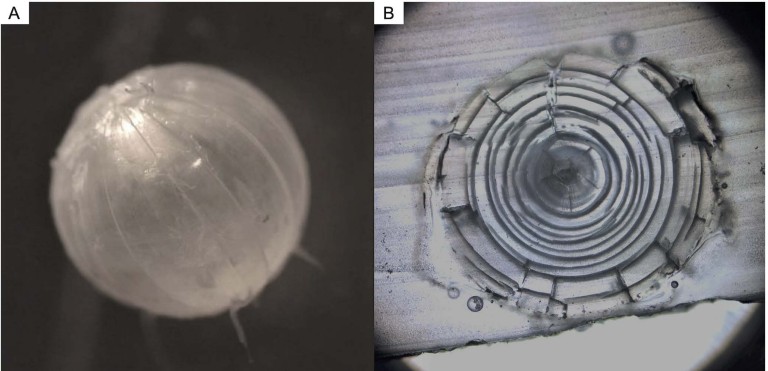

**Fig 2. Pictures of Chinook Salmon (*Oncorhynchus tshawytscha*) eye lenses.** (A) Image of the Chinook Salmon eye lens without the outer cortex taken on a Leica S9i with the accompanying software, LAS X. (B) Cross section image of a Chinook Salmon lens mounted in resin and taken on Image-Pro scientific image analysis software.

isotope-versus-lens position profiles. Although many recent studies have moved toward using lens diameter, presumably to improve reproducibility, none have investigated the researcher-to-researcher variation, nor quantified the gains in reliability that lens diameter measurements provide. This study addresses this existing gap in the literature by examining and quantifying inconsistencies in assigning lamina number among researchers, providing a clearer understanding of the reliability of lens diameter as a metric, and offering recommendations for improving the methodology in this field.

In order to compare the reproducibility between using lamina number and using diameter method for delamination, two researchers each worked on one of two lenses from the same fish without communication. A total of 10 fish were used for this study. Lamina number, diameter (mm) and mass (mg) were assigned, measured, and recorded as metrics to assess

consistency and reproducibility between researchers. In total, 527 lamina samples were generated and submitted for δ¹³C and δ¹⁵N isotope analysis. These results allowed for comparison of isotopic trends across parameters (lamina number and lens diameter) and between individual fish. This study strives to offer a refined and reproducible method for lens delamination by reducing the intrinsic human differences in the delamination process.

## Materials and methods

### Ethics statement

This research was conducted in accordance with protocols approved by the University of California, Davis Institutional Animal Care and Use Committee (IACUC Protocol No: 20979) and authorized under Scientific Collecting Permit (#S-201120001-20240-001). Tissues used in this study were collected from carcasses (i.e., already deceased) of adult Chinook Salmon encountered during routine carcass surveys conducted by the California Department of Fish and Wildlife (CDFW) and the U.S Fish and Wildlife Service (USFWS).

### Sample selection

All specimens were hatchery-origin adult fall and winter-run Chinook Salmon from 2019, identified by the absence of their adipose fins. Individuals were selected from surplus samples remaining after completion of prior analyses. Five adult fall-run and five adult winter-run Chinook Salmon were chosen at random.

### Lens technique

Lenses were extracted by creating an incision in the eyes and then removed with a pair of forceps. Each lens was then stored in a 1.5 mL Eppendorf tube and frozen at −22 °C until delamination.

Both researchers independently delaminated one lens (left or right) from each of the 10 fish, ensuring paired lenses from the same fish were analyzed separately. Lens delamination followed previously established methods [2,8]. The outer cortex was observed on the surface of most lenses and carefully removed by rolling them on clean Kimwipes. The lens was positioned with one of the poles facing upward, where the laminae visibly converge (Fig 3A), and individual laminae were removed using a petal-to-petal motion, starting from the pole, with intermittent application of deionized water to facilitate separation (Fig 3B). Once fully separated, each lamina was often divided into four quadrants (Fig 3C) and then gathered for storage (Fig 3D).

After each lamina was removed, the lens was measured to the nearest thousandth of a millimeter using a Leica S9i imaging microscope with LAS X v5.0.3 imaging software. The process was repeated until the lens core (~0.3 mm) was reached. In this study, the "core" refers to the central, hardened portion of the lens from which no additional laminae could be removed without compromising structural integrity. This region is structurally distinct from the surrounding laminae and lacks further separable layers. It is analogous to what has been described in previous studies as the nucleus [14], the central core [9] or the embryonic core [6,15] / central hardened lens [6]. All laminae were stored in pre-weighed 8x5mm tin capsules (Elemental Microanalysis pressed tin capsules) and air-dried at room temperature in a closed-lid tray for over 24 hours. After drying, samples were weighed to the nearest hundredth of a milligram on a Mettler Toledo Semi-Micro Balance (MS105DU). Individual lamina that exceeded the target weight (> 6.0 mg) were homogenized and subsampled to meet the 0.6–6 mg dried weight range criteria established by the CAMAS Stable Isotope Laboratory at the Idaho State University for δ¹³C and δ¹⁵N analysis. Individual researchers worked independently to ensure their methods remained unbiased.

### Statistical analysis

To quantify the patterns and differences between researchers, a two-sample t-test was used to examine the difference in the average number of laminae generated per person in R [16]. Assumptions of normality was tested using the Shapiro-Wilk test and equality of variances was assessed using Levene's Test in R under the car package. Both assumptions were

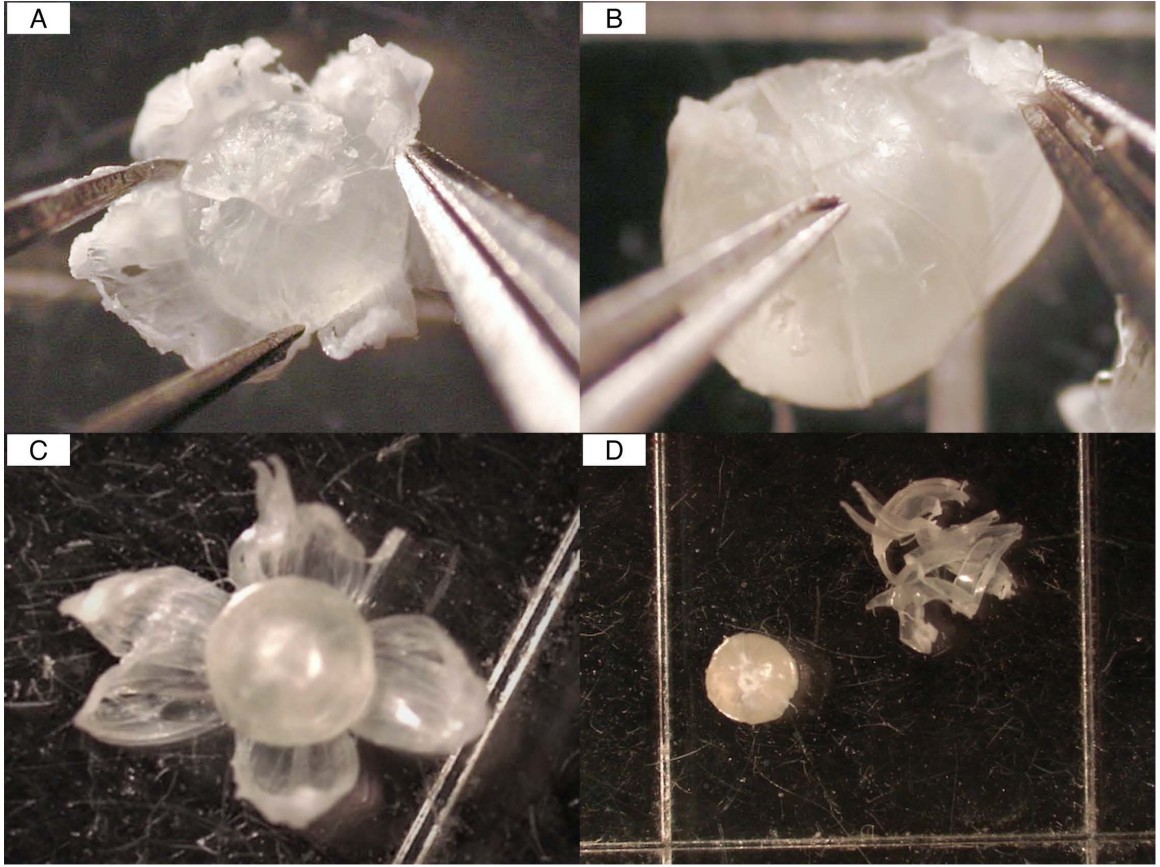

**Fig 3. Overview of lens delamination process.** (A) Lens positioned pole-up for peeling. (B) Lamina removed using a petal-to-petal motion with intermittent hydration. (C) Separated lamina divided into quadrants. (D) Lamina material gathered for storage and SIA submission.

met (S1 Table). Differences of lamina width between the two researchers were then assessed, under the assumption of even growth in the left and right eye from the same fish. The average width of each lamina was calculated by subtracting the lens diameters measured before and after the lamina was removed. To quantify the differences observed, a linear regression model including an interaction term between lamina number and researcher was fitted. Model assumptions were evaluated using Q-Q plots and residuals versus fitted plots (S2 Fig). Mild tail deviations were observed in the Q-Q plot for normality. However, given the large sample size (n = 525), the linear model is considered appropriate [17]. Due to detected heteroscedasticity, robust standard errors HC3 were applied in R using the sandwich package. Significance level of $P < 0.05$ was used to determine the significance of researchers in the differences of lens diameter at the identical lamina number. Patterns in lamina counts, diameter, and isotope values were compared visually between readers using the dplyr and ggplot2 packages in R [16]. The four fish with the largest discrepancy in lamina number between researchers were used to display the variance in data.

## Results

### Between-researcher variation

Researchers 1 and 2 displayed distinct variations in their peeling methods. Notably, Researcher 1 exhibited a tendency to peel thicker laminae compared to Researcher 2, resulting in a mean total lamina count of 24 (+/- 2.1), while Researcher

2, who tended to peel thinner laminae, achieved a higher mean total lamina count of 27.7 (+/- 2.7) (see Figs 4 and 5). Researcher 1's total lamina count did not exceed 26, whereas Researcher 2 acquired a maximum of 33 laminae. A t-test found there to be a statistically significant difference in the average lamina number produced per eye between the two researchers (t = 3.33, P < 0.05).

Researcher 1 consistently recorded larger lens diameters than Researcher 2 for each of the identical laminae numbers. Similar to larger diameters, researcher 1 removed laminae with greater mass than researcher 2 for the same lamina

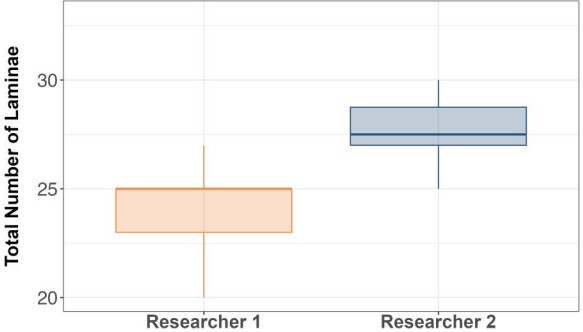

**Fig 4. Box plot displaying differences in total laminae between researchers.** The box denotes the median and interquartile range, and the whiskers extend to 1.5 times the interquartile range.

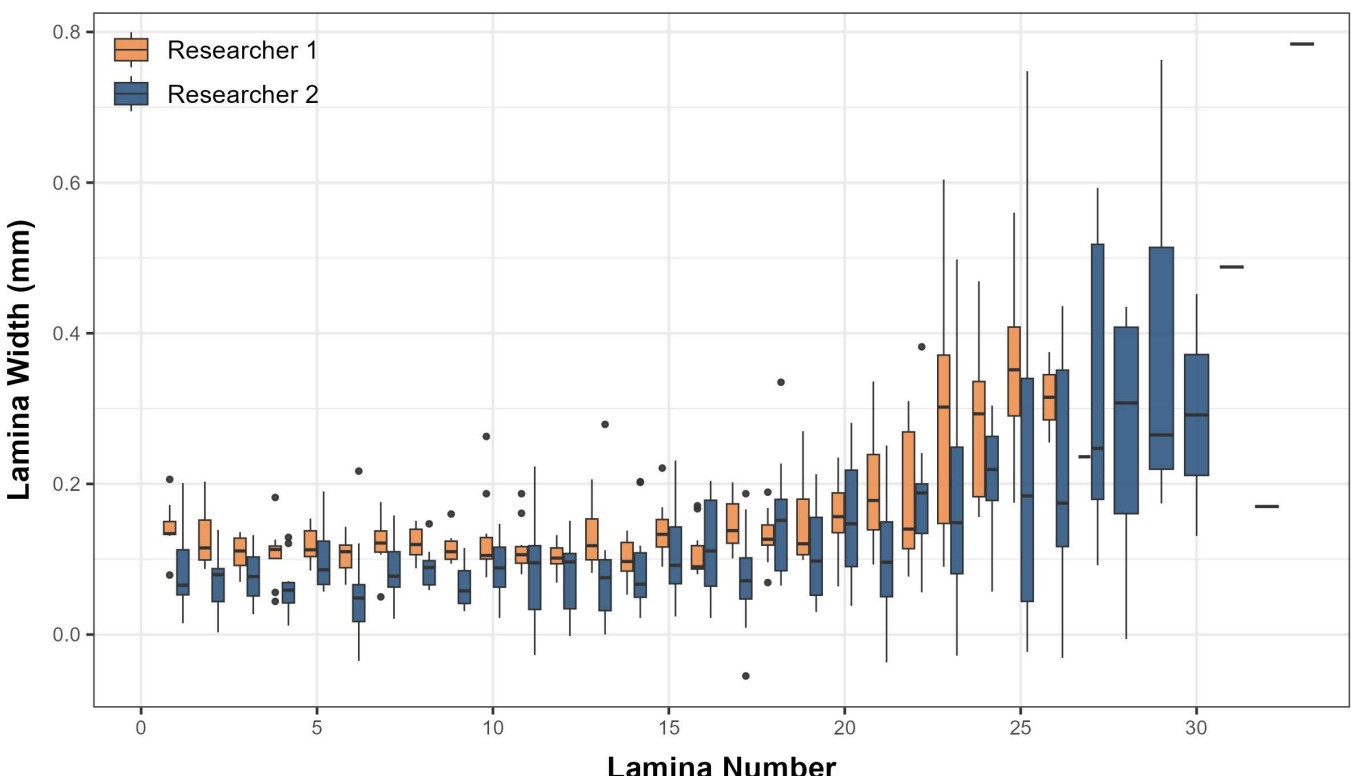

**Fig 5. Box plot showing average lamina widths for each lamina between researchers.** The box denotes the median and interquartile range, and the whiskers denote 1.5 times the interquartile range.

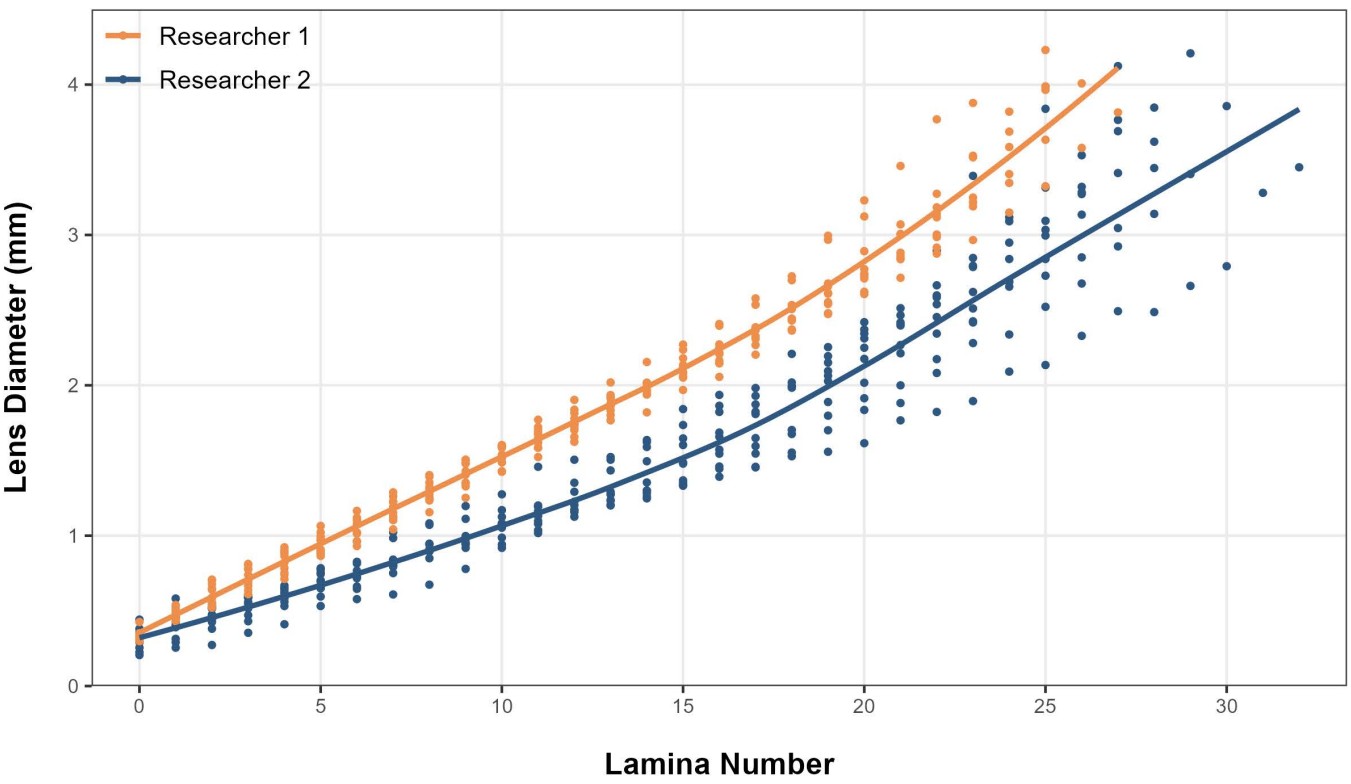

**Fig 6. Scatter plot of lamina number plotted against the lens diameter (mm) by researcher.**

number on average (S1 Fig). Linear regression analysis using heteroscedasticity-consistent (HC3) robust standard errors revealed that both researcher and lamina number, as well as their interaction, significantly influenced lens diameter (S2 Table). Due to between researcher variation linked with lamina number assignment, the same lamina number might correspond to different lens diameters for fish delaminated by different researchers, thus causing a mismatch with isotope values (Fig 6).

**Using isotope values for validation.** A visual comparison was then performed using δ¹³C and δ¹⁵N values from two lenses of the same fish. These plots compared the utilization of assigned lamina numbers and measured lens diameters as metrics, as shown in Figs 7a–d. The δ¹³C and δ¹⁵N values served as grounding references for assessing the relative accuracy between lamina number and lens diameter. When plotting the δ¹³C and δ¹⁵N trajectories against lens diameter, a much more consistent alignment between the two researchers was evident (Figs 7c–d). This suggests that the discrepancy between the isotope trajectories is due to the use of lamina number as a metric.

## Discussion

Stable isotope analysis of fish eye lenses has become increasingly popular for interpreting a fish's life history [8], offering insights into diet [2] and habitat use [5,18]. This tool, combined with other endogenous records like otoliths, offers a comprehensive understanding of fish life histories, which is essential for addressing population-level needs. As research in this area grows, the need for standardized methods to ensure reproducibility becomes more crucial. This is particularly important since eye lens isotopes are interpreted and correlated to other metrics, such as fork length [9,13,19] and are used in combination with other endogenous records like otoliths to provide a comprehensive understanding of the species being studied [20].

In the study of otoliths, researchers count annuli to age fish [21]. This method is well-established in otolith micro-chemistry analysis, leading ecologists to naturally gravitate towards applying similar techniques to eye lens analysis [8]. However, unlike otolith analysis, which reveals a full profile of annuli through grinding or sectioning, eye lens analysis involves peeling laminae off the lens, which does not result in a readable profile [2,20]. The count of lamina is more variable than that of annuli since, without reference to the full profile, the definition or boundary between each lamina can be ambiguous. Therefore, caution is recommended when adapting otolith methodologies to eye lens studies.

Plotting lamina number against lens diameter highlights differences in peeling tendencies between researchers (see Fig 6). This discrepancy arises because each researcher's subjective interpretation of where one lamina ends, and another begins can lead to inconsistent results. Although the isotopic trends of δ¹³C and δ¹⁵N are expected to align between eyes, when lamina number is used as the parameter on the x-axis there is a persistent lag in isotopic trajectories between researchers (Figs 7a–b) [8]. Using lamina number to interpret the isotopic profile of these fish makes it challenging to

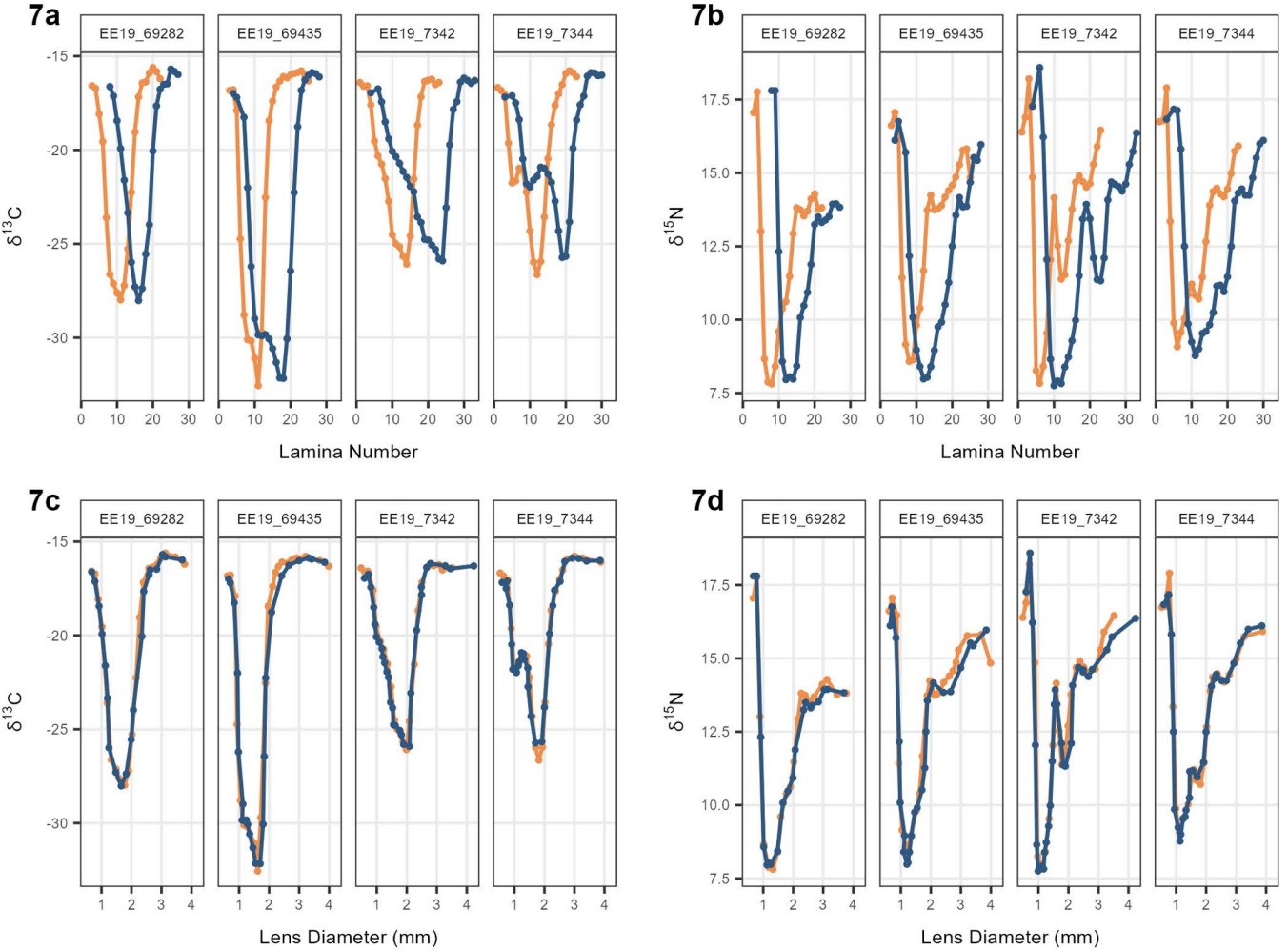

**Fig 7. Plots of δ¹³C and δ¹⁵N values of 4 individuals against lamina number and lens diameter (mm).** (A) δ¹³C values from consecutive laminae plotted against lamina number. (B) δ¹⁵N values from consecutive laminae plotted against lamina number. (C) δ¹⁵N values from consecutive laminae plotted against lens diameter (mm). (D) δ¹⁵N values from consecutive laminae plotted against lens diameter (mm).

extract valuable conclusions when focusing solely on analyzing a specific life stage (see Figs 7a–b). Given these complexities, lens diameter, already adopted in some studies on other teleost species [4,5,8,9], was proposed as an alternative metric for processing and analyzing fish eye lens isotope data. When the δ¹³C and δ¹⁵N values were plotted using lens diameter instead of lamina number, the trends between researchers aligned. This demonstrates that lens diameter is a more reliable and objective parameter for comparing isotope values across laminae. Unlike lamina number, which is subject to individual interpretation and prone to inconsistencies, lens diameter provides a continuous, quantifiable metric that is less affected by user bias. These findings highlight the substantial influence of individual variation in the peeling process and reinforce the need for standardized, objective metrics to improve methodological consistency and quality control.

Despite the differences between researchers, the use of lamina number is still a valid method to answer exploratory questions that can lead to more in-depth research, as it can provide preliminary lifelong diet information when measuring tools are unavailable or too costly. Consistency was observed within researchers, suggesting that lamina number may be used if there is one person consistently delaminating lenses, especially when the study aims to identify shifts in one individual's diet over time. Yet, caution should be taken when comparing results over time and between researchers. Categorizing by lamina number can also be a useful naming system for keeping individual lamina organized for sample submission, storage, and in a database. Delaminating generates many samples per individual fish; having a consistent naming system, such as lamina number is important for data organization.

Overall, lens diameter offers a more dependable marker for growth categorization, overcoming the limitations associated with the use of lamina number. The adoption of lens diameter corrects for biases introduced by human effects that are tied to lamina number. Using lens diameter instead of lamina number as a comparison metric enhances the reliability and reproducibility of lens delamination. Additionally, it allows for the evaluation of results between different studies, offering a universal baseline for comparison.

## Future works

The outer cortex is observed on the surface of most lenses and represents a partially formed, living lamina that has not yet compressed into the hardened crystalline structure that the rest of the lens exhibits [ 6, 7, 22]. Obtaining an accurate measurement of the outer cortex's diameter is challenging due to its irregular shape and uneven edges, which may result from the extraction or freezing process. For the purposes of this methodology paper, the outer cortexes were carefully removed before the lenses were measured by rolling them on clean Kimwipes.

In our group, we have experimented with two approaches to quantify the outer cortex: (1) measuring its widest portion, which proved inconsistent due to the irregular structure once the lens capsule is compromised (due to dissection or thawing), and (2) weighing the dried cortex, which introduced high variability and standard errors due to difficulty in consistently recovering all tissue. While neither method has proven reliable, we believe that refining the lens extraction process, particularly when working with frozen tissue, could improve preservation of the lens capsule and enable more standardized outer cortex measurement. Continuing development of outer cortex measurement is a valuable endeavor that would support more accurate reconstructions of growth history and contribute to establishing a robust relationship between fork length and lens diameter.

## Supporting information

**S1 Fig. Mass (mg) of each lamina plotted against lamina number.** Laminae were weighed using a microbalance. Researcher 1 (orange) peeled fewer and heavier lamina on average. Researcher 2, however, tended to peel lighter lamina, resulting in more lamina.
(TIFF)

**S1 Table. Assumption testing for the two-sample t-test comparing lamina counts between researchers.** Summary of the assumption tests for the two-sample t-test comparing lamina count between researchers. Non-significant p values (p > 0.05) suggest normal distribution within each group and equality of variance between groups, satisfying the assumptions for a standard t-test.
(PDF)

**S2 Fig. Diagnostic plots evaluating assumptions for the linear regression model of lens diameter as a function of lamina number and researcher.** (a) Q-Q plot of residuals colored by researcher. Mild deviations from normality were observed at the tails. (b) Residuals versus fitted values colored by researcher. An increase in variance was observed across fitted values, indicative of heteroscedasticity and confirmed by the Breusch-Pagan test (BP = 71.931, p < 0.001).
(TIFF)

**S2 Table. Linear regression model results for lens diameter as a function of lamina number and researcher. Researcher 2 was the reference group.** Robust standard errors (HC3) were applied to correct for heteroscedasticity. Significant predictors are indicated by p-values, with *** denoting p < 0.001.
(PDF)

**S1 File. Eye lens peeling notes.** Supplemental information on methods.
(PDF)

## Acknowledgments

We extend our gratitude to Paloma Herrera-Thomas for her exceptional illustration of Chinook Salmon and its lens! We are honored to have her illustration in this paper. Additionally, we would like to thank the CAMAS Stable Isotope Laboratory at Idaho State University for conducting the isotope analyses on all our samples and an anonymous reviewer for their comments has helped greatly improve the early version of this manuscript.

## Author contributions

**Conceptualization:** Alexandra Chu, Danhong Ally Li, Miranda Bell-Tilcock, Carson Jeffres, Rachel C. Johnson.

**Data curation:** Alexandra Chu, Danhong Ally Li, Miranda Bell-Tilcock.

**Formal analysis:** Alexandra Chu, Danhong Ally Li, Miranda Bell-Tilcock.

**Funding acquisition:** Carson Jeffres, Rachel C. Johnson.

**Investigation:** Alexandra Chu, Danhong Ally Li.

**Methodology:** Miranda Bell-Tilcock, Carson Jeffres, Rachel C. Johnson.

**Project administration:** Miranda Bell-Tilcock, Carson Jeffres.

**Resources:** Carson Jeffres, Rachel C. Johnson.

**Software:** Alexandra Chu, Danhong Ally Li.

**Supervision:** Miranda Bell-Tilcock.

**Validation:** Miranda Bell-Tilcock, Carson Jeffres, Rachel C. Johnson.

**Visualization:** Alexandra Chu, Danhong Ally Li, Miranda Lowe-Webb.

**Writing – original draft:** Alexandra Chu, Danhong Ally Li.

**Writing – review & editing:** Miranda Bell-Tilcock, Miranda Lowe-Webb, Carson Jeffres, Rachel C. Johnson.

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
