## [Decision Letter · Decision Letter 0]

Dear Dr. Li,

Thank you for submitting your manuscript to PLOS ONE. After careful consideration, we feel that it has merit but does not fully meet PLOS ONE’s publication criteria as it currently stands. Therefore, we invite you to submit a revised version of the manuscript that addresses the points raised during the review process.

We look forward to receiving your revised manuscript.

Kind regards,

Vitor Hugo Rodrigues Paiva, Ph.D.

Academic Editor

PLOS ONE

**Journal Requirements:**

1. When submitting your revision, we need you to address these additional requirements. Please ensure that your manuscript meets PLOS ONE's style requirements, including those for file naming. The PLOS ONE style templates can be found at https://journals.plos.org/plosone/s/file?id=wjVg/PLOSOne_formatting_sample_main_body.pdf and https://journals.plos.org/plosone/s/file?id=ba62/PLOSOne_formatting_sample_title_authors_affiliations.pdf 2. Thank you for stating the following financial disclosure: Funding for this project was provided by California Department of Fish and Wildlife Proposition 1 agreement number P1896030. PIs: Carson Jeffres and Rachel Johnson.   Please state what role the funders took in the study.  If the funders had no role, please state: "The funders had no role in study design, data collection and analysis, decision to publish, or preparation of the manuscript." If this statement is not correct you must amend it as needed. Please include this amended Role of Funder statement in your cover letter; we will change the online submission form on your behalf. 3. When completing the data availability statement of the submission form, you indicated that you will make your data available on acceptance. We strongly recommend all authors decide on a data sharing plan before acceptance, as the process can be lengthy and hold up publication timelines. Please note that, though access restrictions are acceptable now, your entire data will need to be made freely accessible if your manuscript is accepted for publication. This policy applies to all data except where public deposition would breach compliance with the protocol approved by your research ethics board. If you are unable to adhere to our open data policy, please kindly revise your statement to explain your reasoning and we will seek the editor's input on an exemption. Please be assured that, once you have provided your new statement, the assessment of your exemption will not hold up the peer review process. 4. Please include your full ethics statement in the ‘Methods’ section of your manuscript file. In your statement, please include the full name of the IRB or ethics committee who approved or waived your study, as well as whether or not you obtained informed written or verbal consent. If consent was waived for your study, please include this information in your statement as well. 5. Please include captions for your Supporting Information files at the end of your manuscript, and update any in-text citations to match accordingly. Please see our Supporting Information guidelines for more information: http://journals.plos.org/plosone/s/supporting-information.

Reviewers' comments:

Reviewer's Responses to Questions

**Comments to the Author**

1. Is the manuscript technically sound, and do the data support the conclusions?

Reviewer #1: Yes

Reviewer #2: Yes

Reviewer #3: Partly

2. Has the statistical analysis been performed appropriately and rigorously?

Reviewer #1: Yes

Reviewer #2: Yes

Reviewer #3: Yes

3. Have the authors made all data underlying the findings in their manuscript fully available?

Reviewer #1: Yes

Reviewer #2: Yes

Reviewer #3: Yes

4. Is the manuscript presented in an intelligible fashion and written in standard English?

Reviewer #1: Yes

Reviewer #2: Yes

Reviewer #3: Yes

**Reviewer #1:**  In this article, the author presents a method to maximize the reproducibility of stable isotope diet chronologies using fish eye lenses. The content of this paper is comprehensive and the data is detailed and reliable. However, there is one question that needs to be addressed before publication.

For this study, the researchers utilized adult Chinook Salmon (Oncorhynchus tshawytscha) to conduct the investigation and found that the lens diameter in relation to isotope patterns was nearly identical. However, has this result been validated in other fish species? I suggest that the authors provide data from an additional species of fish.

**Reviewer #2:**  Review of PONE-D-24-57172: “Enhancing reproducibility in stable isotope analysis (SIA) of fish eye lenses: A comparison between lamina number and diameter”.

General comment:

The manuscript presents a well-structured study comparing two methods for analysing fish eye lens using stable isotope analysis (SIA). The study is relevant and timely, given the increasing use of eye lenses for dietary and life-history reconstructions. The methodology is clear, and the results convincingly demonstrate that using lens diameter instead of lamina number improves reproducibility. Its findings have practical implications for standardising ecological research. However, some areas require some clarification, additional justification, or minor revisions.

Minor comments:

Line 20: The sentence "Fish eye lenses grow sequentially throughout their ontogeny, resulting in a structure of multiple layers, or laminae" is accurate but slightly awkward. Suggestion: Rephrase to "Fish eye lenses grow throughout their ontogeny, forming multiple sequential layers, or laminae," to improve readability.

Line 22: The analogy "much like tree rings" is useful but lacks context for readers unfamiliar with SIA. Suggestion: Add a brief clarification, e.g., "much like tree rings, which record environmental conditions over time."

Line 26: The phrasing "peeled lenses from each eye of the same adult Chinook Salmon" is unclear about the experimental design. Suggestion: Clarify as "Each researcher independently delaminated one lens from each of 10 adult Chinook Salmon."

Line 33: "Analyzing the laminae based on lamina number resulted in significant variability between researchers" lacks specificity. Suggestion: Specify the nature of variability, e.g., "Analysis based on lamina number showed significant variability in isotope values and lamina counts between researchers."

Line 40: The opening sentence is maybe a bit abrupt. Suggestion: Start with a broader context, e.g., "Understanding animal diets is fundamental to ecology, informing conservation and management strategies, yet capturing lifetime dietary shifts remains challenging."

Lines 40-47: This first paragraph could use some references.

Line 48: The claim about eye lenses as an "emerging archival tissue" is true and could be elaborated. Suggestion: Add "due to their metabolically inert laminae, which preserve stable isotope values reflective of diet over time."

Line 65: “or isotope maps” instead of “or geographic map”.

Line 68: The example of the Atlantic goliath grouper is relevant but long/wordy. Suggestion: "For example, SIA of eye lenses revealed habitat shifts in Atlantic goliath grouper (Epinephelus itajara), emphasizing the role of mangrove connectivity in conservation [11]."

Line 81: Microscopes’ cameras do provide good resolution to measure eye lens diameters. However, are they always needed if there is no access to costly microscope cameras? The use of a digital calliper (that is not as expensive as a microscope’s camera) is also a valuable tool to measure eye lens diameters.

Line 102: Fish origin is not mentioned. Add context if available.

Line 105: What temperature were the Eppendorf tubes kept at?

Line 106: The delamination process is a bit unclear regarding lens assignment. Suggestion: Revise to "Both researcher independently delaminated one lens (left or right) from each of the 10 fish, ensuring paired lenses from the same fish were analyzed separately."

Line 107: Citation accuracy should be verified. Suggestion: Ensure [6] directly supports the delamination protocol or if another reference related to delamination could be added. Additional, reverse citing order i.e., [5], then [6].

Lines 107-108 and Lines 238-239: “A membrane of gelatinous material, often referred to as the outer cortex” has already been defined in the introduction. You can start this sentence by: “The outer cortex…”

Line 112: how were the laminae dried? What temperature, for how long?

Line 116: “meet the 0.6mg-6 mg” 6 mg seems a lot for stable isotope measurements. Is this correct?

Line 121: Figure 3 and its description are well presented. This is a very valuable section. However, most of the caption description could be incorporated into the main text.Line 129: T-tests are suitable for comparing the means of two independent groups (the researchers). However, there is no statement about the fact that the data are normally distributed, and variances are equal, which is a prerequisite to use these tests.

Line 174: “Figures 7a, 7b, 7C, and 7d” could be changed to (Figs 7a–d) for clarity.

Line 186: no need to redefine SIA.

Line 193: “and are used in combination with other endogenous records like otoliths to provide a comprehensive understanding of the species being studied” – any references here?

Line 195: From there, the whole discussion section could use some references. For instance, citing the studies who are already using eye lens diameter.

Line 210: As comment above, lens diameter is presented as a novel proposal, though prior studies used it. Suggestion: Adjust to "Lens diameter, already adopted in some studies [references], proved more reliable than lamina number."

Line 237: Future Works - This section is brief and relevant. Any thoughts on developing a standardised method to measure the outer cortex?

Line 252: in the methods, it reads that the stable isotope lab used was: “All samples were then submitted to the CAMAS Stable Isotope Laboratory at Idaho State University for δ¹³C and δ¹⁵N analysis” – which is it?

**Reviewer #3:**  Congratulations on your work. You have relevant research that contributes to the development of SRI and optimizes resources for work with fish. I think you have a great piece of writing, however, my main comment is that in many sections of your writing, you repeat concepts and omit the rationale for what you want to express by not citing many sections, primarily in the introduction, methodology, and discussion. While it is true that we can recognize many general patterns, there are statements that necessarily need to be substantiated. Furthermore, I believe the use of tables in your writing is excessive, and in many cases, they could be omitted and better summarized in the text, or simply not presented at all. Always consider presenting only what is essential and helps facilitate understanding of your writing. Finally, and respectfully, I would like to suggest that you may want to reconsider submitting this work as an article, as the content is limited in terms of the breadth of information and the sample size (species and individuals). I believe you should consult similar works to what you propose and base your proposal on them. I once again congratulate you on the excellent work you have produced.

**Do you want your identity to be public for this peer review?** For information about this choice, including consent withdrawal, please see our Privacy Policy

Reviewer #1: No

Reviewer #2: No

Reviewer #3: No

---

## [Author Response · Author response to Decision Letter 1]

12 May 2025

Authors’ response to the Reviewers

We thank the Academic Editor and all three reviewers for their thoughtful and constructive feedback on our manuscript. We have addressed all comments carefully and have revised the manuscript accordingly. Below we provide a detailed, point-by-point response. Reviewer comments are shown in italic, followed by our responses highlighted in blue.

Comments to the Authors

Reviewer #1: In this article, the author presents a method to maximize the reproducibility of stable isotope diet chronologies using fish eye lenses. The content of this paper is comprehensive and the data is detailed and reliable. However, there is one question that needs to be addressed before publication.

For this study, the researchers utilized adult Chinook Salmon (Oncorhynchus tshawytscha) to conduct the investigation and found that the lens diameter in relation to isotope patterns was nearly identical. However, has this result been validated in other fish species? I suggest that the authors provide data from an additional species of fish.

Response: Thank you for this thoughtful suggestion. While our study focused specifically on adult Chinook Salmon (Oncorhynchus tshawytscha), we note that the reproducibility of lens diameter as a metric has already been shown across several other teleost species, including white grunt (Haemulon plumierii) (Wallace et al., 2014) and Clear Lake hitch (Lavinia exilicauda chi) (Young et al., 2022), among others.

To address this, we revised the Discussion section and think it strengthens the broader context of our results:

"Given these complexities, lens diameter, already adopted in some studies on other teleost species (Curtis et al., 2020; Quaeck-Davies et al., 2018; Wallace et al., 2014; Young et al., 2022), was proposed as an alternative metric for processing and analyzing fish eye lens isotope data."

Reviewer #2: Review of PONE-D-24-57172: “Enhancing reproducibility in stable isotope analysis (SIA) of fish eye lenses: A comparison between lamina number and diameter”.

General comment:

The manuscript presents a well-structured study comparing two methods for analysing fish eye lens using stable isotope analysis (SIA). The study is relevant and timely, given the increasing use of eye lenses for dietary and life-history reconstructions. The methodology is clear, and the results convincingly demonstrate that using lens diameter instead of lamina number improves reproducibility. Its findings have practical implications for standardising ecological research. However, some areas require some clarification, additional justification, or minor revisions.

Response: We appreciate the careful and constructive suggestions and have carefully addressed each point in detail below:

Minor comments:

Line 20: The sentence "Fish eye lenses grow sequentially throughout their ontogeny, resulting in a structure of multiple layers, or laminae" is accurate but slightly awkward. Suggestion: Rephrase to "Fish eye lenses grow throughout their ontogeny, forming multiple sequential layers, or laminae," to improve readability.

Response: We accept the suggested phrasing, as it improves readability and preserves the key messages.

Line 22: The analogy "much like tree rings" is useful but lacks context for readers unfamiliar with SIA. Suggestion: Add a brief clarification, e.g., "much like tree rings, which record environmental conditions over time."

Response: We appreciate this suggestion and have incorporated context to make the analogy clearer for a broader audience.

Line 26: The phrasing "peeled lenses from each eye of the same adult Chinook Salmon" is unclear about the experimental design. Suggestion: Clarify as "Each researcher independently delaminated one lens from each of 10 adult Chinook Salmon."

Response: We have revised the sentence as suggested to clarify the experimental design.

Line 33: "Analyzing the laminae based on lamina number resulted in significant variability between researchers" lacks specificity. Suggestion: Specify the nature of variability, e.g., "Analysis based on lamina number showed significant variability in isotope values and lamina counts between researchers."

Response: We agree and have updated the sentence to specify the nature of variability observed with the lamina number approach.

Line 40: The opening sentence is maybe a bit abrupt. Suggestion: Start with a broader context, e.g., "Understanding animal diets is fundamental to ecology, informing conservation and management strategies, yet capturing lifetime dietary shifts remains challenging."

Response: We agree that opening with a broader context improves the flow and have revised the sentence accordingly. While we did not use the suggested phrasing verbatim, we incorporated the idea that understanding animal diets is fundamental to ecology. The point regarding the challenges of capturing lifetime dietary shifts was already included later in the paragraph and remains unchanged.

Lines 40-47: This first paragraph could use some references.

Response: Thank you for bringing this to our attention. We added additional relevant citations to the first paragraph!

Line 48: The claim about eye lenses as an "emerging archival tissue" is true and could be elaborated. Suggestion: Add "due to their metabolically inert laminae, which preserve stable isotope values reflective of diet over time."

Response: While we had previously elaborated on the metabolically inert nature of eye lens laminae and their use in diet reconstruction in the following sentences, we have now incorporated the latter part of your suggested phrasing (“which preserve stable isotope values reflective of diet over time”) to improve clarity.

Line 65: “or isotope maps” instead of “or geographic map”.

Response: We have edited the sentence and replaced “geographic map” with “spatial map” to better define “isoscape”..

Line 68: The example of the Atlantic goliath grouper is relevant but long/wordy. Suggestion: "For example, SIA of eye lenses revealed habitat shifts in Atlantic goliath grouper (Epinephelus itajara), emphasizing the role of mangrove connectivity in conservation [11]."

Response: This is a great suggestion - we’ve replaced the previous sentence with the reviewer's suggestion.

Line 81: Microscopes’ cameras do provide good resolution to measure eye lens diameters. However, are they always needed if there is no access to costly microscope cameras? The use of a digital calliper (that is not as expensive as a microscope’s camera) is also a valuable tool to measure eye lens diameters.

Response: We added references of other studies that use alternative (less costly) measuring tools (i.e. ocular micrometers or digital/Vernier calipers). We appreciate this suggestion, as it makes this method more globally accessible.

Line 102: Fish origin is not mentioned. Add context if available.

Response: We added on the fish origin under the Methods section.

Line 105: What temperature were the Eppendorf tubes kept at?

Response: They are kept in a freezer set to -22 degree Celsius and we’ve added this information in the manuscript.

Line 106: The delamination process is a bit unclear regarding lens assignment. Suggestion: Revise to "Both researchers independently delaminated one lens (left or right) from each of the 10 fish, ensuring paired lenses from the same fish were analyzed separately."

Response: We updated the phrasing to match what was suggested.

Line 107: Citation accuracy should be verified. Suggestion: Ensure [6] directly supports the delamination protocol or if another reference related to delamination could be added. Additional, reverse citing order i.e., [5], then [6].

Response: Upon re-evaluating Reference [6], we agree that it does not directly support our statement regarding the delamination protocol. We have therefore removed this citation to maintain accuracy and clarity.

Lines 107-108 and Lines 238-239: “A membrane of gelatinous material, often referred to as the outer cortex” has already been defined in the introduction. You can start this sentence by: “The outer cortex…”

Response: Thank you for pointing this out. To avoid redundancy and improve flow, we revised the sentence at Lines 238–239 to begin with “The outer cortex…” and rephrased it for clarity. The updated sentence now reads: “The outer cortex is observed on the surface of most lenses and represents a partially formed, living lamina that has not yet compressed into the hardened crystalline structure exhibited by the rest of the lens.”

Line 112: how were the laminae dried? What temperature, for how long?

Response: We have clarified the drying method in the revised text to specify that laminae were air-dried at room temperature in a closed-lid tray for over 24 hours.

Line 116: “meet the 0.6mg-6 mg” 6 mg seems a lot for stable isotope measurements. Is this correct?

Response: We checked with the isotope facility at Idaho State University and yes - it is the limit set by them for our samples based on their calibration range. We have clarified the sentence.

Line 121: Figure 3 and its description are well presented. This is a very valuable section. However, most of the caption description could be incorporated into the main text.

Response: That is a great suggestion- thank you. In response, we have integrated the text from Figure 3 caption into the main narrative of the Methods section to improve flow and clarity. We retained a simplified version of the caption to complement the figure while minimizing redundancy.

Line 129: T-tests are suitable for comparing the means of two independent groups (the researchers). However, there is no statement about the fact that the data are normally distributed, and variances are equal, which is a prerequisite to use these tests.

Response: We added language and reporting to address the assumptions associated with our statistical tests. Our data met the assumption of normality and homogeneity of variance for the t-test and we added the results of the Shapiro-Wilk tests for normality and Levene’s tests for homogeneity of variance in the Statistical analysis section and in the Supplemental (S1 Table). We modified our original ANOVA-based analysis of lens diameter variation by lamina number and researcher to better reflect the continuous nature of lamina number as a predictor and the interaction we aimed to test. We transitioned to a linear regression framework with an interaction term, which allowed us to assess whether the relationship between lamina number and lens diameter differed by researcher. As part of the model evaluation, we assessed normality and homoscedasticity using Q-Q plots and residuals versus fitted plots (S2 Figure), which indicated significant heteroscedasticity. To address this, we applied heteroscedasticity-consistent (HC3) robust standard errors. Our conclusions did not change, but the revised model results are reported in the Supplemental (Table S2), and we added clarifying language in the Methods section. We appreciate the reviewer’s suggestion, which improved the rigor and clarity of our statistical analysis.

Line 174: “Figures 7a, 7b, 7C, and 7d” could be changed to (Figs 7a–d) for clarity.

Response: Accepted suggestion.

Line 186: no need to redefine SIA.

Response: Accepted suggestion.

Line 193: “and are used in combination with other endogenous records like otoliths to provide a comprehensive understanding of the species being studied” – any references here?

Response: Added citation to this sentence here. Citing work by Stounberg, J. et. al, 2022, where eye lens and otoliths are both analyzed for trace elements to establish a baseline correlation between the two for inquiry of comprehensive life history of fish.

Line 195: From there, the whole discussion section could use some references. For instance, citing the studies who are already using eye lens diameter.

Response: Relevant citations added throughout Discussion.

Line 210: As comment above, lens diameter is presented as a novel proposal, though prior studies used it. Suggestion: Adjust to "Lens diameter, already adopted in some studies [references], proved more reliable than lamina number."

Response: Accepted suggestion.

Line 237: Future Works - This section is brief and relevant. Any thoughts on developing a standardised method to measure the outer cortex?

Response: Thank you for your comment and that is a great question. We have attempted with other projects to measure the outer cortex, but no consistent results could be generated from those attempts. I personally think the process of lens extraction from frozen lens tissue can be perfected or experimented on to reduce damage and contact with the lens during the process to protect integrity of the lens capsule so standard measurements can be relied on but we have yet have the resources and opportunity to investigate. We have expanded the Future works section to include details and thoughts from our prior efforts, and it reads:

“In our group, we have experimented with two approaches to quantify the outer cortex: (1) measuring its widest portion, which proved inconsistent due to the irregular structure once the lens capsule is compromised, and (2) weighing the dried cortex, which introduced high variability and standard errors due to difficulty in consistently recovering all tissue. While neither method has proven reliable, we believe that refining the lens extraction process, particularly when working with frozen tissue, could improve preservation of the lens capsule and enable more standardized outer cortex measurement. Continuing development of outer cortex measurement is a valuable endeavor that would support more accurate reconstructions of growth history and contribute to establishing a robust relationship between fork length and lens diameter.”

Line 252: in the methods, it reads that the stable isotope lab used was: “All samples were then submitted to the CAMAS Stable Isotope Laboratory at Idaho State University for δ¹³C and δ¹⁵N analysis” – which is it?

Response: Thank you so much for pointing out the discrepancy. We changed the mentioning all to the ISU facility.

Reviewer #3: Congratulations on your work. You have relevant research that contributes to the development of SRI and optimizes resources for work with fish. I think you have a great piece of writing, however, my main comment is that in many sections of your writing, you repeat concepts and omit the rationale for what you want to express by not citing many sections, primarily in the introduction, methodology, and discussion. While it is true that we can recognize many general patterns, there are statements that necessarily need to be substantiated.

Response: In response to this helpful suggestion, we carefully revised the Introduction, Methods, and Discussion sections to improve clarity, remove redundancies, and ensure that key statements are fully substantiated with appropriate citations.

Furthermore, I believe the use of tables in your writing is excessive, and in many cases, they could be omitted and better summarized in the text, or simply not presented at all. Always consider presenting only what is essential and helps facilitate understanding of your writing.

Response: Thank you for this helpful advice. We carefully reviewed all figures and tables to ensure that only essential information is presented. We were unable to identify any data tables in the main manuscript that may have prompted this comment. However, we suspect that you may be referring to Figure 3, which is a multi-panel image (Fig. 3A–D) showing the delamination procedure. In response, we revised the main text to summarize these steps more clearly and shortened the figure caption. We believe that including this visual sequence provides valuable guidance for readers and helps clarify the eye lens processing method.

Finally, and respectfully, I would like to suggest that you may want to reconsider submitting this work as an article, as the content is limited in terms of the breadth of information and the sample size (species and individuals). I believe you should consult similar works to what you propose and base your proposal on them.

Response: Thank you for this comment. To clarify, the manuscript was submitted as a Research Article in accordance

---

## [Decision Letter · Decision Letter 1]

Enhancing reproducibility in stable isotope analysis (SIA) of fish eye lenses: A comparison between lamina number and diameter

PONE-D-24-57172R1

Dear Dr. Li,

We’re pleased to inform you that your manuscript has been judged scientifically suitable for publication and will be formally accepted for publication once it meets all outstanding technical requirements.

Kind regards,

Vitor Hugo Rodrigues Paiva, Ph.D.

Academic Editor

PLOS ONE

Additional Editor Comments (optional):

Reviewers' comments:

Reviewer's Responses to Questions

**Comments to the Author**

Reviewer #1: All comments have been addressed

Reviewer #2: All comments have been addressed

Reviewer #3: All comments have been addressed

2. Is the manuscript technically sound, and do the data support the conclusions?

Reviewer #1: Yes

Reviewer #2: (No Response)

Reviewer #3: Yes

3. Has the statistical analysis been performed appropriately and rigorously?

Reviewer #1: Yes

Reviewer #2: (No Response)

Reviewer #3: Yes

4. Have the authors made all data underlying the findings in their manuscript fully available?

Reviewer #1: Yes

Reviewer #2: (No Response)

Reviewer #3: Yes

5. Is the manuscript presented in an intelligible fashion and written in standard English?

Reviewer #1: Yes

Reviewer #2: (No Response)

Reviewer #3: Yes

Reviewer #1: I believe the authors have addressed all the concerns, so I recommend that the article be accepted for publication.

Reviewer #2: (No Response)

Reviewer #3: Thank you for the version you've included. I've noticed improvements in both its structure and content. I still believe you have a relevant topic worth sharing with the scientific community. I have no major comments on the work, and I believe it can be accepted.

**Do you want your identity to be public for this peer review?** For information about this choice, including consent withdrawal, please see our Privacy Policy

Reviewer #1: No

Reviewer #2: No

Reviewer #3: No

---

## [Editor Report · Acceptance letter]

PONE-D-24-57172R1

PLOS ONE

Dear Dr. Li,

I'm pleased to inform you that your manuscript has been deemed suitable for publication in PLOS ONE. Congratulations! Your manuscript is now being handed over to our production team.

Kind regards,

on behalf of

Dr. Vitor Hugo Rodrigues Paiva

Academic Editor

PLOS ONE